# Conditional Antibody Design as 3D Equivariant Graph Translation

**Xiangzhe Kong**[1]   **Wenbing Huang**[4,5,*] **Yang Liu**[1,2,3,*]

[1]Dept. of Comp. Sci. & Tech., Institute for AI, BNRist Center, Tsinghua University
[2]Institute for AI Industry Research (AIR), Tsinghua University
[3]Beijing Academy of Artificial Intelligence
[4]Gaoling School of Artificial Intelligence, Renmin University of China
[5] Beijing Key Laboratory of Big Data Management and Analysis Methods, Beijing, China
`Jackie_KXZ@outlook.com, hwenbing@126.com, liuyang2011@tsinghua.edu.cn`

## Abstract

Antibody design is valuable for therapeutic usage and biological research. Existing deep-learning-based methods encounter several key issues: 1) incomplete context for Complementarity-Determining Regions (CDRs) generation; 2) incapability of capturing the entire 3D geometry of the input structure; 3) inefficient prediction of the CDR sequences in an autoregressive manner. In this paper, we propose Multi-channel Equivariant Attention Network (MEAN) to co-design 1D sequences and 3D structures of CDRs. To be specific, MEAN formulates antibody design as a conditional graph translation problem by importing extra components including the target antigen and the light chain of the antibody. Then, MEAN resorts to E(3)-equivariant message passing along with a proposed attention mechanism to better capture the geometrical correlation between different components. Finally, it outputs both the 1D sequences and 3D structure via a multi-round progressive full-shot scheme, which enjoys more efficiency and precision against previous autoregressive approaches. Our method significantly surpasses state-of-the-art models in sequence and structure modeling, antigen-binding CDR design, and binding affinity optimization. Specifically, the relative improvement to baselines is about 23% in antigen-binding CDR design and 34% for affinity optimization.

## 1 Introduction

Antibodies are Y-shaped proteins used by our immune system to capture specific pathogens. They show great potential in therapeutic usage and biological research for their strong specificity: each type of antibody usually binds to a unique kind of protein that is called antigen (Basu et al., 2019). The binding areas are mainly located at the so-called Complementarity-Determining Regions (CDRs) in antibodies (Kuroda et al., 2012). Therefore, the critical problem of antibody design is to identify CDRs that bind to a given antigen with desirable properties like high affinity and colloidal stability (Tiller & Tessier, 2015). There have been unremitting efforts made for antibody design by using deep generative models (Saka et al., 2021; Jin et al., 2021). Traditional methods focus on modeling only the 1D CDR sequences, while a recent work (Jin et al., 2021) proposes to co-design the 1D sequences and 3D structures via Graph Neural Network (GNN).

Despite the fruitful progress, existing approaches are still weak in modeling the spatial interaction between antibodies and antigens. For one thing, the context information is insufficiently considered. The works (Liu et al., 2020; Jin et al., 2021) only characterize the relation between CDRs and the backbone context of the same antibody chain, without the involvement of the target antigen and other antibody chains, which could lack complete clues to reflect certain important properties for antibody design, such as binding affinity. For another, they are still incapable of capturing the entire 3D geometry of the input structures. One vital property of the 3D Biology is that each structure (molecular, protein, etc) should be independent to the observation view, exhibiting E(3)-equivariance[1]. To fulfill

---

*Corresponding authors: Wenbing Huang, Yang Liu.

[1]E(3) is the group of Euclidean transformations: rotations, reflections, and translations.

this constraint, the method by Jin et al. (2021) pre-processes the 3D coordinates as certain invariant features before feeding them to the model. However, such pre-procession will lose the message of direction in the feature and hidden spaces, making it less effective in characterizing the spatial proximity between different residues in antibodies/antigens. Further, current generative models (Saka et al., 2021; Jin et al., 2021) predict the amino acids one by one; such autoregressive fashion suffers from low efficiency and accumulated errors during inference.

To address the above issues, this paper formulates antibody design as E(3)-equivariant graph translation, which is equipped with the following contributions: **1. New Task.** We consider *conditional generation*, where the input contains not only the heavy-chain context of CDRs but also the information of the antigen and the light chain. **2. Novel Model.** We put forward an end-to-end *Multi-channel Equivariant Attention Network* (MEAN) that outputs both 1D sequence and 3D structure of CDRs. MEAN operates directly in the space of 3D coordinates with E(3)-equivariance (other than previously-used E(3)-invariant models), such that it maintains the full geometry of residues. By alternating between an internal context encoder and an external attentive encoder, MEAN leverages 3D message passing along with equivariant attention mechanism to capture long-range and spatially-complicated interactions between different components of the input complex. **3. Efficient Prediction.** Upon MEAN, we propose to progressively generate CDRs over multiple rounds, where each round updates both the sequences and structures in a full-shot manner. This *progressive full-shot decoding* strategy is less prone to accumulated errors and more efficient during the inference stage, compared with traditional autoregressive models (Gebauer et al., 2019; Jin et al., 2021). We validate the efficacy of our model on three challenging tasks: sequence and structure modeling, antigen-binding CDR design and binding affinity optimization. Compared to previous methods that neglect the context of light chain and antigen (Saka et al., 2021; Akbar et al., 2022; Jin et al., 2021), our model achieves significant improvement on modeling the 1D/3D joint distribution, and makes a great stride forward on recovering or optimizing the CDRs that bind to the target antigen.

## 2 RELATED WORK

**Antibody Design** Early approaches optimize antibodies with hand-crafted energy functions (Li et al., 2014; Lapidoth et al., 2015; Adolf-Bryfogle et al., 2018), which rely on costly simulations and have intrinsic defects because the inter-chain interactions are complicated in nature and cannot be fully captured by simple force fields or statistical functions (Graves et al., 2020). Hence, attention has been paid to applying deep generative models for 1D sequence prediction (Alley et al., 2019; Liu et al., 2020; Saka et al., 2021; Akbar et al., 2022). Recently, to further involve the 3D structure, Jin et al. (2021) proposes to design the sequences and structures of CDRs simultaneously. It represents antibodies with E(3)-invariant features upon the distance matrix and generates the sequences autoregressively. Its follow-up work (Jin et al., 2022) further involves the epitope, but only considers CDR-H3 without all other components in the antibody. Different from the above learning-based works, our method considers a more complete context by importing the 1D/3D information of the antigen and the light chain. More importantly, we develop an E(3)-equivariant model that is skilled in representing the geometry of and the interactions between 3D structures. Note that antibody design assumes both the CDR sequences and structures are unknown, whereas general protein design predicts sequences based on structures (Ingraham et al., 2019; Karimi et al., 2020; Cao et al., 2021).

**Equivariant Graph Neural Networks** The growing availability of 3D structural data in various fields (Jumper et al., 2021) leads to the emergence of geometrically equivariant graph neural networks (Klicpera et al., 2020; Liu et al., 2021; Puny et al., 2021; Han et al., 2022). In this paper, we exploit the scalarization-based E(n)-equivariant GNNs (Satorras et al., 2021) as the building block of our MEAN. Specifically, we adopt the multichannel extension by Huang et al. (2022) that naturally complies with the multichannel representation of a residue. Furthermore, we have developed a novel equivariant attention mechanism in MEAN to better capture antibody-antigen interactions.

## 3 METHOD

### 3.1 PRELIMINARIES, NOTATIONS AND TASK FORMULATION

An antibody is a Y-shaped protein (Figure 1) with two symmetric sets of chains, each composed of a heavy chain and a light chain (Kuroda et al., 2012). In each chain, there are some constant

domains, and a *variable domain* ($V_H/V_L$) that has three *Complementarity Determining Regions (CDRs)*. Antigen-binding sites occur on the variable domain where the interacting regions are mostly CDRs, especially CDR-H3. The remainder of the variable domain other than CDRs is structurally well conserved and often called *framework region* (Kuroda et al., 2012; Jin et al., 2021). Therefore, previous works usually formalize the antibody design problem as finding CDRs that fit into the given framework region (Shin et al., 2021; Akbar et al., 2022). As suggested by (Fischman & Ofran, 2018; Jin et al., 2021), we focus on generating CDRs in heavy chains since they contribute the most to antigen-binding affinity and are the most challenging ones to characterize. Nevertheless, in contrast to previous studies, we additionally incorporate the antigen and the light chain into the context in the form of antibody-antigen complexes, to better control the binding specificity of generated antibodies.

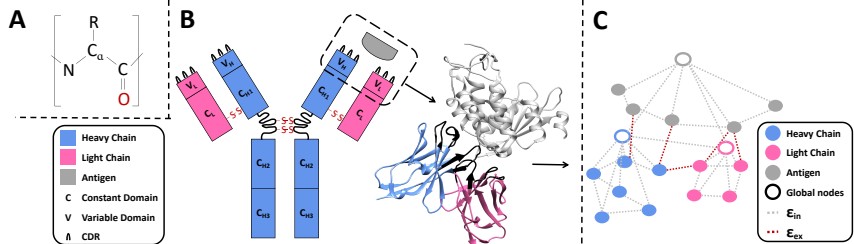

Figure 1: (**A**) The structure of a residue, where the backbone atoms we use are $N, C_\alpha, C, O$. (**B**) The structure of an antibody which is symmetric and Y-shaped, and we focus on the three versatile CDRs on the variable domain of the heavy chain. (**C**) Schematic graph construction for the antigen-antibody complex, with global nodes, internal context edges $\mathcal{E}_{in}$ and external interaction edges $\mathcal{E}_{ex}$.

We represent each antibody-antigen complex as a graph of three spatially aggregated components, denoted as $\mathcal{G} = (\mathcal{V} := \{\mathcal{V}_H, \mathcal{V}_L, \mathcal{V}_A\}, \mathcal{E} := \{\mathcal{E}_{in}, \mathcal{E}_{ex}\})$. Here, the components $\mathcal{V}_H, \mathcal{V}_L, \mathcal{V}_A$ correspond to the nodes (*i.e.* the residues) of the heavy chain, the light chain and the antigen, respectively; $\mathcal{E}_{in}$ and $\mathcal{E}_{ex}$ separately contain internal edges within each component and external edges across components. To be specific, each node in $\mathcal{V}$, *i.e.*, $v_i = (h_i, \mathbf{Z}_i)$ is represented as a trainable feature embedding vector $h_i = s_{a_i} \in \mathbb{R}^{d_a}$ according to its amino acid type $a_i$ and a matrix of coordinates $\mathbf{Z}_i \in \mathbb{R}^{3 \times m}$ consisting of $m$ backbone atoms. In our case, we set $m = 4$ by choosing 4 backbone atoms $\{N, C_\alpha, C, O\}$, where $C_\alpha$ denotes the alpha carbon of the residue and others refer to the atoms composing the peptide bond (Figure 1). We denote the residues in CDRs to be generated as $\mathcal{V}_C = \{v_{c_1}, v_{c_2}, ..., v_{c_{n(c)}}\}$, which is a subset of $\mathcal{V}_H$. Since the information of each $v_{c_i}$ is unknown in the first place, we initialized its input feature with a mask vector and the coordinates according to the even distribution between the residue right before CDRs (namely, $v_{c_1-1}$) and the one right after CDRs (namely, $v_{c_{n(c)}+1}$). In our main experiments (§ 4), we select the 48 residues of the antigen closest to the antibody in terms of the $C_\alpha$ distance as the epitope like Jin et al. (2021). Instead of such hand-crafted residue selection, in Appendix L, we also incorporate the full antigen and let our method determine the epitope automatically, where the efficacy of our method is still exhibited.

**Edge construction** We now detail how to construct the edges. For the internal edges, $\mathcal{E}_{in}$ is defined as the edges connecting each pair of nodes within the same component if the spatial distance in terms of $C_\alpha$ is below a cutoff distance $c_1$. Note that adjacent residues in a chain are spatially close and we always include the edge between adjacent residues in $\mathcal{E}_{in}$. In addition, we assign distinct edge types by setting $e_{ij} = 1$ for those adjacency residues and $e_{ij} = 0$ for others, to incorporate the 1D position information. For the external edges $\mathcal{E}_{ex}$, they are derived if the nodes from two different components have a distance less than a cutoff $c_2$ ($c_2 > c_1$). It is indeed necessary to separate internal and external interactions because their distance scales are very different. The external connections actually represent the interface between different chains, which dominates the binding affinity (Chakrabarti & Janin, 2002), and they are formed mainly through inter-molecular forces instead of chemical bonds that form the internal connections within chains (Yan et al., 2008). Note that all edges are constructed without the information of ground-truth CDR positions.

**Global nodes** The shape of CDR loops is closely related to the conformation of the framework region (Baran et al., 2017). Therefore, to make the generated CDRs aware of the entire context of the chain they are in, we additionally insert a global node into each component, by connecting it to all other nodes in the component. Besides, the global nodes of different components are linked to

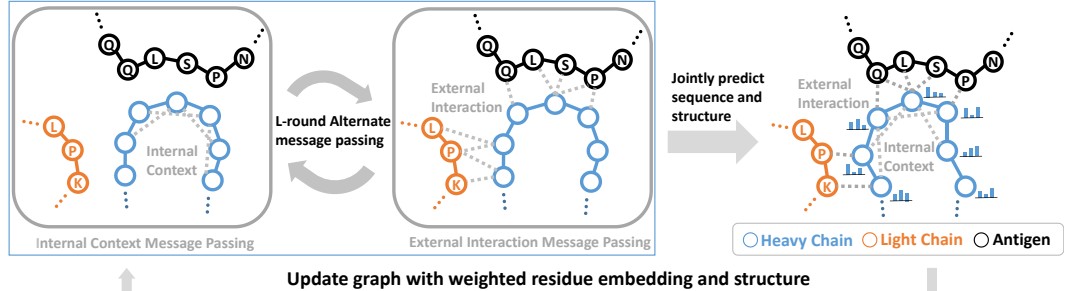

Figure 2: The overview of MEAN and the progressive full-shot decoding. In each iteration, we alternate the internal context encoding and external interaction encoding over $L$ layers, and then update the input features and coordinates of CDRs for the next iteration with the predicted values.

each other, and all edges induced by the global nodes are included in $\mathcal{E}_{\text{in}}$. The coordinates of a global node are given by the mean of all coordinates of the variable domain of the corresponding chains.

**Task formulation** Given the 3D antibody-antigen complex graph $\mathcal{G} = (\mathcal{V}, \mathcal{E})$, we seek a 3D equivariant translator $f$ to generate the amino acid type and 3D conformation for each residue in CDRs $\mathcal{V}_C$. Distinct from 1D sequence translation in conventional antibody design, our task requires to output 3D information, and more importantly, we emphasize equivariance to reflect the symmetry of our 3D world—the output of $f$ will translate/rotate/reflect in the same way as its input. We now present how to design $f$ in what follows.

## 3.2 MEAN: MULTI-CHANNEL EQUIVARIANT ATTENTION NETWORK

To derive an effective translator, it is crucial to capture the 3D interactions of the residues in different chains. The message passing mechanism in $E(3)$-equivariant GNNs (Satorras et al., 2021; Huang et al., 2022) will fulfil this purpose. Particularly, we develop the Multi-channel Equivariant Attention Network (MEAN) to characterize the geometry and topology of the input antibody-antigen complex. Each layer of MEAN alternates between the two modules: internal context encoder and external interaction encoder, which is motivated by the biological insight that the external interactions between antibodies and antigens are different from those internal interactions within each heavy/light chain. After several layers of the message passing, the node representations and coordinates are transformed into the predictions by an output module. Notably, all modules are $E(3)$-equivariant.

**Internal context encoder** Similar to GMN (Huang et al., 2022), we extend EGNN (Satorras et al., 2021) from one single input vector to multichannel coordinates, since each residue is naturally represented by multiple backbone atoms. Suppose in layer $l$ the node features are $\{h_i^{(l)} | i = 1, 2, ..., n\}$ and the coordinates are $\{\boldsymbol{Z}_i^{(l)} | i = 1, 2, ..., n\}$. We denote the relative coordinates between node $i$ and $j$ as $\boldsymbol{Z}_{ij}^{(l)} = \boldsymbol{Z}_i^{(l)} - \boldsymbol{Z}_j^{(l)}$. Then, the information of each node is updated in the following form.

$$m_{ij} = \phi_m(h_i^{(l)}, h_j^{(l)}, \frac{(\boldsymbol{Z}_{ij}^{(l)})^\top \boldsymbol{Z}_{ij}^{(l)}}{\|(\boldsymbol{Z}_{ij}^{(l)})^\top \boldsymbol{Z}_{ij}^{(l)}\|_F}, e_{ij}), \tag{1}$$

$$h_i^{(l+0.5)} = \phi_h(h_i^{(l)}, \sum_{j \in \mathcal{N}(i|\mathcal{E}_{\text{in}})} m_{ij}), \tag{2}$$

$$\boldsymbol{Z}_i^{(l+0.5)} = \boldsymbol{Z}_i^{(l)} + \frac{1}{|\mathcal{N}(i|\mathcal{E}_{\text{in}})|} \sum_{j \in \mathcal{N}(i|\mathcal{E}_{\text{in}})} \boldsymbol{Z}_{ij}^{(l)} \phi_Z(m_{ij}), \tag{3}$$

where $\mathcal{N}(i|\mathcal{E}_{\text{in}})$ denotes the neighbors of node $i$ regarding the internal connections $\mathcal{E}_{\text{in}}$, $\| \cdot \|_F$ returns the Frobenius norm, and $\phi_m, \phi_x, \phi_h, \phi_Z$ are all Multi-Layer Perceptons (MLPs) (Gardner & Dorling, 1998). Basically, $m_{ij}$ gathers the $E(3)$-invariant messages from all neighbors; then it is used to update $h_i$ via $\phi_h$ and $\boldsymbol{Z}_i$ via $\phi_Z$ that is additionally left multiplied with $\boldsymbol{Z}_{ij}^{(l)}$ to keep the direction information. As $\mathcal{E}_{\text{in}}$ also contains the connections between global nodes in different components, the encoder here actually involves inter-component message passing, although in a global sense. We use the superscript $(l + 0.5)$ to indicate the features and coordinates that will be further updated by the external attentive encoder in this layer.

**External attentive encoder** This module exploits the graph attention mechanism (Veličković et al., 2017) to better describe the correlation between the residues of different components, but different from Veličković et al. (2017), we design a novel $E(3)$-equivariant graph attention scheme based on the multichannel scalarization in the above internal context encoder. Formally, we have:

$$\alpha_{ij} = \frac{\exp(q_i^\top k_{ij})}{\sum_{j \in \mathcal{N}(i|\mathcal{E}_{\text{ex}})} \exp(q_i^\top k_{ij})}, \tag{4}$$

$$h_i^{(l+1)} = h_i^{(l+0.5)} + \sum_{j \in \mathcal{N}(i|\mathcal{E}_{\text{ex}})} \alpha_{ij} v_{ij}, \tag{5}$$

$$\boldsymbol{Z}_i^{(l+1)} = \boldsymbol{Z}_i^{(l+0.5)} + \sum_{j \in \mathcal{N}(i|\mathcal{E}_{\text{ex}})} \alpha_{ij} \boldsymbol{Z}_{ij}^{(l+0.5)} \phi_Z(v_{ij}), \tag{6}$$

where, $\mathcal{N}(i|\mathcal{E}_{\text{ex}})$ denotes the neighbors of node $i$ defined by the external interactions $\mathcal{E}_{\text{ex}}$; $q_i$, $k_{ij}$, and $v_{ij}$ are the query, key, and value vectors, respectively, and $\alpha_{ij}$ is the attention weight from node $j$ to $i$. Specifically, $q_i = \phi_q(h_i^{(l+0.5)})$, $k_{ij} = \phi_k\big(\frac{(\boldsymbol{Z}_{ij}^{(l+0.5)})^\top \boldsymbol{Z}_{ij}^{(l+0.5)}}{\|(\boldsymbol{Z}_{ij}^{(l+0.5)})^\top \boldsymbol{Z}_{ij}^{(l+0.5)}\|_F}, h_j^{(l+0.5)}\big)$, and $v_{ij} = \phi_v\big(\frac{(\boldsymbol{Z}_{ij}^{(l+0.5)})^\top \boldsymbol{Z}_{ij}^{(l+0.5)}}{\|(\boldsymbol{Z}_{ij}^{(l+0.5)})^\top \boldsymbol{Z}_{ij}^{(l+0.5)}\|_F}, h_j^{(l+0.5)}\big)$ are all $E(3)$-invariant, where the functions $\phi_q, \phi_k, \phi_v$ are MLPs.

**Output module** After $L$ layers of the alternations between the last two modules, we further conduct Eq. (1-3) to output the hidden feature $\tilde{h}_i$ and coordinates $\tilde{\boldsymbol{Z}}_i$. To predict the probability of each amino acid type, we apply a SoftMax on $\tilde{h}_i$: $p_i = \text{Softmax}(\tilde{h}_i)$, where $p_i \in \mathbb{R}^{n_a}$ is the predicted distribution over all amino acid categories.

A desirable property of MEAN is that it is $E(3)$-equivariant. We summarize it as a formal theorem below, with the proof deferred to Appendix E.

**Theorem 1.** *We denote the translation process by MEAN as $\{(p_i, \tilde{\boldsymbol{Z}}_i)\}_{i \in \mathcal{V}_C} = f(\{h_i^{(0)}, \boldsymbol{Z}_i^{(0)}\}_{i \in \mathcal{V}})$, then $f$ is $E(3)$-equivariant. In other words, for each transformation $g \in E(3)$, we have $\{(p_i, g \cdot \tilde{\boldsymbol{Z}}_i)\}_{i \in \mathcal{V}_C} = f(\{h_i^{(0)}, g \cdot \boldsymbol{Z}_i^{(0)}\}_{i \in \mathcal{V}})$, where the group action $\cdot$ is instantiated as $g \cdot \boldsymbol{Z} := \boldsymbol{O}\boldsymbol{Z}$ for orthogonal transformation $\boldsymbol{O} \in \mathbb{R}^{3 \times 3}$ and $g \cdot \boldsymbol{Z} := \boldsymbol{Z} + \boldsymbol{t}$ for translation transformation $\boldsymbol{t} \in \mathbb{R}^3$.*

### 3.3 PROGRESSIVE FULL-SHOT DECODING

Traditional methods (such as RefineGNN (Jin et al., 2021)) unravel the CDR sequence in an autoregressive way: generating one amino acid at one time. While such strategy is able to reduce the generation complexity, it inevitably incurs expensive computing and memory overhead, and will hinder the training owing to the vanishing gradient for long CDR sequences. It also acumulates errors during the inference stage. Here, thanks to the rich expressivity of MEAN, we progressively generate the CDRs over $T$ iterations ($T$ is much smaller than the length of the CDR sequences), and in each iteration, we predict the amino acid type and 3D coordinates of all the nodes in $\mathcal{V}_C$ at once. We call our scheme as *full-shot decoding* to distinguish it from previous autoregressive approaches.

To be specific, given the CDRs' amino acid distribution and conformation $\{p_i^{(t)}, \tilde{\boldsymbol{Z}}_i^{(t)}\}_{i \in \mathcal{V}_C}$ from iteration $t$, we first update the embeddings of all nodes: $h_i' = \sum_{j=1}^{n_a} p_i^{(t)}(j) s_j, \forall i \in \mathcal{V}_C$, where $p_i^{(t)}(j)$ returns the probability for the $j$-th class and $s_j$ is the corresponding learnable embedding as defined before. Such weighted strategy leads to less accumulated error during inference compared to the maximum selection counterpart. We then replace the CDRs with the new values $\{h_i', \tilde{\boldsymbol{Z}}_i^{(t)}\}_{i \in \mathcal{V}_C}$, and denote the new graph as $\mathcal{G}^{(t+1)}$. The edges are also constructed dynamically according to the new graph. We update the next iteration as $\{p_i^{(t+1)}, \tilde{\boldsymbol{Z}}_i^{(t+1)}\}_{i \in \mathcal{V}_C} = \text{MEAN}(\mathcal{G}^{(t+1)})$.

For sequence prediction, we exert supervision for each node at each iteration:

$$\mathcal{L}_{\text{seq}} = \frac{1}{T} \sum_t \frac{1}{|\mathcal{V}_C|} \sum_{i \in \mathcal{V}_C} \ell_{ce}(p_i^{(t)}, \hat{p}_i), \tag{7}$$

where $\ell_{ce}$ denotes the cross entropy between the predicted distribution $p_i^{(t)}$ and the true one $\hat{p}_i$.

For structure prediction, we only exert supervision on the output iteration. Since there are usually noises in the coordination data, we adopt the Huber loss (Huber, 1992) other than the common MSE

loss to avoid numerical instability (further explanation can be found in Appendix F):

$$\mathcal{L}_{\text{struct}} = \frac{1}{|\mathcal{V}_C|} \sum_{i \in \mathcal{V}_C} \ell_{\text{huber}}(\tilde{\boldsymbol{Z}}_i^{(T)}, \hat{\boldsymbol{Z}}_i), \tag{8}$$

where $\hat{\boldsymbol{Z}}_i$ is the label. One benefit of the structure loss is that it conducts directly in the coordinate space, and it is still $E(3)$-invariant as our model is $E(3)$-equivariant. This is far more efficient than the loss function used in RefineGNN (Jin et al., 2021). To ensure invariance, RefineGNN should calculate over pairwise distance and angle other than coordinates, which is tedious but necessary since it can only perceive the input of node and edge features after certain invariant transformations. Finally, we balance the above two losses with $\lambda$ to form $\mathcal{L} = \mathcal{L}_{\text{seq}} + \lambda \mathcal{L}_{\text{struct}}$.

## 4 EXPERIMENTS

We assess our model on the three challenging tasks: **1.** The generative task on the Structural Antibody Database (Dunbar et al., 2014) in § 4.1; **2.** Antigen-binding CDR-H3 design from a curated benchmark of 60 diverse antibody-antigen complexes (Adolf-Bryfogle et al., 2018) in § 4.2; **3.** Antigen-antibody binding affinity optimization on Structural Kinetic and Energetic database of Mutant Protein Interactions (Jankauskaitė et al., 2019) in § 4.3. We also present a promising pipeline to apply our model in scenarios where the binding position is unknown in § 4.4.

Four baselines are selected for comparison. For the first one, we use the **LSTM**-based approach by Saka et al. (2021); Akbar et al. (2022) to encode the context of the heavy chain and another LSTM to decode the CDRs. We implement the cross attention between the encoder and the decoder, but utilize the sequence information only. Built upon LSTM, we further test **C-LSTM** to consider the entire context of the antibody-antigen complex, where each component is separated by a special token. **RefineGNN** (Jin et al., 2021) is related to our method as it also considers the 3D geometry for antibody generation, but distinct from our method it is only $E(3)$-invariant and autoregressively generate the amino acid type of each residue. Since its original version only models the heavy chain, we extend it by accommodating the whole antibody-antigen complex, which is denoted as **C-RefineGNN**; concretely, each component is identified with a special token in the sequence and a dummy node in the structure. As denoted before, we term our model as **MEAN**. We train each model for 20 epochs and select the checkpoint with the lowest loss on the validation set for testing. We use the Adam optimizer with the learning rate 0.001. For MEAN, we run 3 iterations for the progressive full-shot decoding. More details are provided in Appendix D. For LSTM and RefineGNN, we borrow their default settings and source codes for fair comparisons.

### 4.1 SEQUENCE AND STRUCTURE MODELING

For quantitative evaluation, we employ Amino Acid Recovery (AAR), defined as the overlapping rate between the predicted 1D sequences and ground truths, and Root Mean Square Deviation (RMSD) regarding the 3D predicted structure of CDRs. Thanks to its inherent equivariance, our model can directly calculate RMSD of coordinates, unlike other baselines that resort to the Kabsch technique Kabsch (1976) to align the predicted and true coordinates prior to the RMSD computation. Our model requires each input complex to be complete (consisting of heavy chains, light chains, and antigens). Hence, we choose 3,127 complexes from the Structural Antibody Database (Dunbar et al., 2014, SAbDab) and remove other illegal datapoints that lack light chain or antigen. All selected complexes are renumbered under the IMGT scheme (Lefranc et al., 2003). As suggested by Jin et al. (2021), we split the dataset into training, validation, and test sets according to the clustering of CDRs to maintain the generalization test. In detail, for each type of CDRs, we first cluster the sequences via MMseqs2 (Steinegger & Söding, 2017) that assigns the antibodies with CDR sequence identity above 40% to the same cluster, where the BLOSUM62 substitution matrix (Henikoff & Henikoff, 1992) is adopted to calculate the sequence identity. The total numbers of clusters for CDR-H1, CDR-H2, and CDR-H3 are 765, 1093, and 1659, respectively. Then we split all clusters into training, validation, and test sets with a ratio of 8:1:1. We conduct 10-fold cross validation to obtain reliable results. Further details are provided in Appendix A.

**Results** Table 1 (Top) demonstrates that our MEAN significantly surpasses all other methods in terms of both the 1D sequence and 3D structure modeling, which verifies the effectiveness of MEAN

Table 1: Top: 10-fold cross validation mean (standard deviation) for 1D sequence and 3D structure modeling on SAbDab (§4.1). Bottom: evaluations under the setting of Jin et al. (2021), denoted with a superscript ∗.

| Model | CDR-H1 | | CDR-H2 | | CDR-H3 | |
|---|---|---|---|---|---|---|
| | AAR | RMSD | AAR | RMSD | AAR | RMSD |
| LSTM | 40.98±5.20% | - | 28.50±1.55% | - | 15.69±0.91% | - |
| C-LSTM | 40.93±5.41% | - | 29.24±1.08% | - | 15.48±1.17% | - |
| RefineGNN | 39.40±5.56% | 3.22±0.29 | 37.06±3.09% | 3.64±0.40 | 21.13±1.59% | 6.00±0.55 |
| C-RefineGNN | 33.19±2.99% | 3.25±0.40 | 33.53±3.23% | 3.69±0.56 | 18.88±1.37% | 6.22±0.59 |
| MEAN | **58.29±7.27%** | **0.98±0.16** | **47.15±3.09%** | **0.95±0.05** | **36.38±3.08%** | **2.21±0.16** |
| LSTM∗ | 28.02% | - | 24.39% | - | 18.92% | - |
| RefineGNN∗ | 30.07% | 0.97 | 27.70% | **0.73** | 27.60% | **2.12** |
| MEAN∗ | **62.78%** | **0.94** | **52.04%** | 0.89 | **39.87%** | 2.20 |

in modeling the underlying distribution of the complexes. When comparing LSTM/RefineGNN with C-LSTM/C-RefineGNN, it is observed that further taking the light chain and the antigen into account sometimes leads to even inferior performance. This observation suggests it will cause a negative effect if the interdependence between CDRs and the extra input components is not correctly revealed. On the contrary, MEAN is able to deliver consistent improvement when enriching the context, which will be demonstrated later in § 5. We also compare MEAN with LSTM and RefineGNN on the same split as RefineGNN (Jin et al., 2021), denoted as MEAN∗, LSTM∗, and RefineGNN∗, respectively. Specifically, both LSTM∗ and RefineGNN∗ are trained on the full training set, while our MEAN only accesses its subset composed of complete complexes, which is approximately 52% of the full dataset. All three models are evaluated on the same subset of the test set for fair comparisons. Even so, MEAN still outperforms the other two methods remarkably in terms of AAR and achieves comparable results regarding RMSD in Table 1 (Bottom), validating its strong generalization ability.

## 4.2 ANTIGEN-BINDING CDR-H3 DESIGN

We perform fine-grained validation on designing CDR-H3 that binds to a given antigen. In addition to AAR and RMSD, we further adopt TM-score (Zhang & Skolnick, 2004; Xu & Zhang, 2010), which calculates the global similarity between two protein structures and ranges from 0 to 1, to evaluate how well the CDRs fit into the frameworks. We use the official implementation[2] to calculate TM-score. For autoregressive baselines, we follow Jin et al. (2021) to alleviate the affects of accumulated errors by generating 10,000 CDR-H3s and select the top 100 candidates with the lowest PPL for evaluation. Notably, our MEAN reduces accumulated errors significantly by the progressive full-shot decoding strategy, therefore can directly generate the candidates. We also include RosettaAD (Adolf-Bryfogle et al., 2018), a widely used baseline of conventional approach, for comparison. We benchmark all methods with the 60 diverse complexes carefully selected by Adolf-Bryfogle et al. (2018) (RAbD). The training is still conducted on the SAbDab dataset used in § 4.1, but we eliminate all antibodies from SAbDab whose CDR-H3s share the same cluster as those in RAbD to avoid any potential data leakage. We then divide the remainder into the training and validation sets by a ratio of 9:1. The numbers of clusters/antibodies are 1,443/2,638 for training and 160/339 for validation.

**Results** As shown in Table 2, MEAN outperforms all baselines by a large margin in terms of both AAR and TM-score. Particularly on TM-score, the value by MEAN approaches above 0.99, implying that the designed structure is almost the same as the original one. To better show this, we visualize an example in Figure 3, where the generated fragment by MEAN almost overlaps with the ground truth, while the result of RefineGNN exhibits an apparent bias.

## 4.3 AFFINITY OPTIMIZATION

It is crucial to optimize various properties like binding affinity of antibodies for therapeutic purposes. This can be formulated as a search problem over the intrinsic space of generative models. In our case, we jointly optimize the sequence and structure of CDR-H3 to improve the binding affinity of

---

[2]https://zhanggroup.org/TM-score/

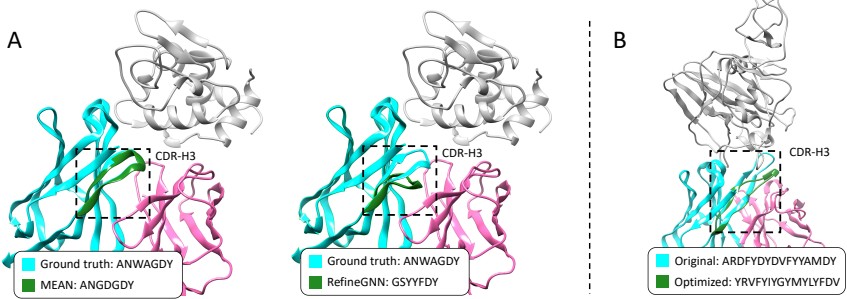

Figure 3: (**A**) The structure of two antigen-binding CDR-H3s (PDB: 1ic7) designed by MEAN (left, RMSD=0.49) and RefineGNN (right, RMSD=3.04). The sequences are also provided in the annotations. (**B**) The antibody-antigen complex structure after affinity optimization (PDB: 2vis, $\Delta\Delta G$ = -5.13), with the original and optimized sequences displayed in the annotations.

any given antibody-antigen complex. For evaluation, we employ the geometric network from Shan et al. (2022) to predict the change in binding energy ($\Delta\Delta G$) after optimization. Particularly, we leverage the official checkpoint[3] that is trained on the Structural Kinetic and Energetic database of Mutant Protein Interactions V2.0 (Jankauskaitė et al., 2019, SKEMPI V2.0). $\Delta\Delta G$ is calibrated under the unit of kcal/mol, and lower $\Delta\Delta G$ indicates better binding affinity. Since all the methods model backbone structures only, we use Rosetta (Alford et al., 2017) to do sidechain packing before affinity prediction. To ensure the expected generalizability, we select a total of 53 antibodies from its training set (i.e. SKEMPI V2.0) for affinity optimization. Besides, we split SAbDab (pre-processed by the strategy in § 4.1) into training and validation sets in a ratio of 9:1 for pretraining the model.

Following Jin et al. (2021), we exploit the Iterative Target Augmentation (Yang et al., 2020, ITA) algorithm to tackle the optimization problem. Since the original algorithm is designed for discrete properties, we adapt it for compatibility with our affinity scorer of continuous values. Please refer to Appendix B for detailed description of the adaption. During the process, we discard any unrealistic candidate with PPL above 10 in accordance with Jin et al. (2021). It is observed that our model learns the constraints of net charge, motifs, and repeating amino acids (Jin et al., 2021) implicitly, therefore we do not need to impose them on our model explicitly.

Table 2: Left: Amino acid recovery (AAR), TM-score and RMSD for CDR-H3 design on RAbD benchmark (§ 4.2). Middle: Average affinity change after optimization textsection 4.3). Right: Average CDR lengths and speedups by our full-shot decoding compared to the iterative-refinement decoding.

| Model | AAR | TM-score | RMSD | | Model | $\Delta\Delta G$ | | length | |
|---|---|---|---|---|---|---|---|---|---|
| RosettaAD | 22.50% | 0.9435 | 5.52 | | Random | +1.52 | | CDR-H1 | 7.9 |
| LSTM | 22.36% | - | - | | LSTM | -1.48 | | CDR-H2 | 7.6 |
| C-LSTM | 22.18% | - | - | | C-LSTM | -1.83 | | CDR-H3 | 14.1 |
| RefineGNN | 29.79% | 0.8303 | 7.55 | | RefineGNN | -3.98 | | speedup (train / infer) | |
| C-RefineGNN | 28.90% | 0.8317 | 7.21 | | C-RefineGNN | -3.79 | | CDR-H1 | 2.1x / 1.5x |
| MEAN | **36.77%** | **0.9812** | **1.81** | | MEAN | **-5.33** | | CDR-H2 | 2.1x / 1.6x |
| | | | | | | | | CDR-H3 | 4.1x / 2.8x |

**Results** As shown in Table 2 (middle), our MEAN models achieve obvious progress towards discovering antibodies with better binding affinity. This further validates the advantage of explicitly modeling the interface with MEAN. Moreover, we provide the predicted $\Delta\Delta G$ of mutating the CDRs to random sequences, denoted as Random, for better interpretation of the results. We provide further interpretation in Appendix C. We also provide a visualization example in Figure 3 (B), which indicates our MEAN does produce a novel CDR-H3 sequence/structure, with improved affinity.

## 4.4 CDR-H3 DESIGN WITH DOCKED TEMPLATE

We further provide a possible pipeline to utilize our model in scenarios when the binding complex is unknown. Specifically, for antigens from RAbD (Adolf-Bryfogle et al., 2018), we aim at generating binding antibodies with high affinity. To this end, we first select an antibody from the database and remove its CDR-H3, then we use HDOCK (Yan et al., 2017) to dock it to the target antigen to obtain a template of antibody-antigen complex. With this template, we employ our model to

---

[3]https://github.com/HeliXonProtein/binding-ddg-predictor

generate antigen-binding CDR-H3s in the same way as § 4.2. To alleviate the risk of docking inaccuracy, we compose 10 such templates for each antigen and retain the highest scoring one in the subsequent generation. We first refine the generated structure with OpenMM (Eastman et al., 2017) and Rosetta (Alford et al., 2017), and then use the energy functions in Rosetta to measure the binding affinity. The comparison of the affinity distribution between the generated antibodies by our method, those by C-RefineGNN, and the original ones in RAbD is shown in Figure 4 (B). Obviously, the antibodies designed by our MEAN exhibit higher predicted binding affinity. We also present a tightly binding example in Figure 4 (A).

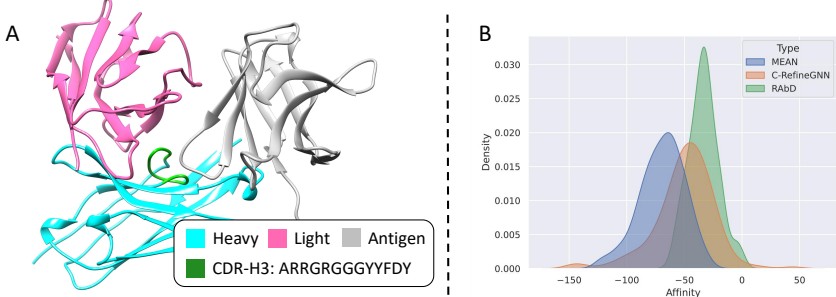

Figure 4: (**A**) CDR-H3 designed by MEAN based on a docked template targeting the antigen from PDB 5b8c (Affinity=-59.6). (**B**) The affinity distribution of the antibodies.

## 5 ANALYSIS

We test if each proposed technique is necessary in MEAN. Table 3 shows that the removal of either the global nodes or the attention mechanism induces performance detriment. This is reasonable since the global nodes transmit information within and between components globally, and the attentive module concentrates on the local information around the interface of different components. In practice, the attentive module also provides interpretability over the significance of pairwise residue interactions, as illustrated in Appendix I. In addition, it is observed that only using the heavy chain weakens the performance apparently, and fails to derive feasible solution for the affinity optimization task, which empirically supports the necessity of inputting antigens and light chains in MEAN. Moreover, we implement a variant of MEAN by replacing the progressive full-shot decoding with the iterative refinement operation used in RefineGNN, whose performance is worse than MEAN. As discussed before, our full-shot decoding is much more efficient than the iterative refinement process, since the number of iterations in MEAN is 3 which is much smaller than that of the refinement-based variant. As reported in Table 2 (right), our method speeds up approximately 2 to 5 times depending on the lengths of the CDR sequences. We also analyze the complexity, MEAN injected with randomness, and the progressive decoding process in Appendix G, H, and J, respectively.

Table 3: Ablations of MEAN.

| Model | SAbDab (CDR-H3) | | RAbD | | | SKEMPI |
|---|---|---|---|---|---|---|
| | AAR | RMSD | AAR | TM-score | RMSD | $\Delta\Delta G$ |
| w/o global node | 33.89±6.81% | 2.62±1.03 | 35.07% | 0.9798 | **1.81** | -3.68 |
| w/o attention | 35.89±3.44% | 2.24±0.31 | 36.56% | 0.9745 | 1.95 | failed |
| heavy chain only | 33.62±6.20% | 2.23±0.17 | 35.82% | 0.9728 | 1.97 | failed |
| iterative refinement | 25.85±1.88% | 3.61±0.75 | 35.19% | 0.9629 | 3.20 | -5.19 |
| MEAN | **36.38±3.08%** | **2.21±0.16** | **36.77%** | **0.9812** | **1.81** | **-5.33** |

## 6 CONCLUSION

In this paper we formulate antibody design as translation from the the entire context of antibody-antigen complex to versatile CDRs. We propose multi-channel equivariant attention network (MEAN) to identify and encode essential local and global information within and between different chains. We also propose progressive full-shot decoding strategy for more efficient and precise generation. Our model outperforms baselines by a large margin in terms of three generation task including distribution learning on 1D sequences and 3D structures, antigen-binding CDR-H3 design, and affinity optimization. Our work presents insights for modeling antibody-antigen interactions in further research.

ACKNOWLEDGMENTS

This work is jointly supported by the Vanke Special Fund for Public Health and Health Discipline Development of Tsinghua University, the National Natural Science Foundation of China (No. 61925601, No. 62006137), Guoqiang Research Institute General Project of Tsinghua University (No. 2021GQG1012), Beijing Academy of Artificial Intelligence, Beijing Outstanding Young Scientist Program (No. BJJWZYJH012019100020098).

REPRODUCIBILITY

The codes for our MEAN are available at https://github.com/THUNLP-MT/MEAN.

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

## A   DETAILS OF SEQUENCE AND STRUCTURE MODELING

**10-fold dataset splits**   We provide the number of clusters and antibodies in each fold of our 10-fold cross validation. When selecting fold $i$ for testing, we use fold $i - 1$ for validation (for fold 1 as the test set, we use fold 10 for validation) and the union of other folds for training.

Table 4: Statistics of each fold of 10-fold cross validation.

|  | CDR-H1 | | CDR-H2 | | CDR-H3 | |
| --- | --- | --- | --- | --- | --- | --- |
|  | cluster | antibody | cluster | antibody | cluster | antibody |
| fold 1 | 77 | 299 | 110 | 237 | 166 | 317 |
| fold 2 | 77 | 260 | 110 | 343 | 166 | 301 |
| fold 3 | 77 | 363 | 110 | 288 | 166 | 348 |
| fold 4 | 77 | 286 | 109 | 315 | 166 | 271 |
| fold 5 | 77 | 195 | 109 | 419 | 166 | 279 |
| fold 6 | 76 | 241 | 109 | 217 | 166 | 345 |
| fold 7 | 76 | 427 | 109 | 266 | 166 | 306 |
| fold 8 | 76 | 326 | 109 | 321 | 166 | 320 |
| fold 9 | 76 | 210 | 109 | 374 | 166 | 309 |
| fold 10 | 76 | 520 | 109 | 347 | 165 | 331 |
| Total | 765 | 3,127 | 1,093 | 3,127 | 1659 | 3,127 |

**RefineGNN paper setting**   In Table 1, we also compare LSTM, RefineGNN, and MEAN on the same split used in the paper of RefineGNN (Jin et al., 2021) and the models are denoted as LSTM*, RefineGNN*, and MEAN*. Here we provide the sizes of training/validation/test set as well as their subsets of complete complexes in Table 5. We use the subsets to train MEAN*, which is only about 52% of the full sets, and all three models are evaluated on the subset of the test set for fair comparison. For RefineGNN*, we directly use the official checkpoints provided by Jin et al. (2021) for evaluation.

Table 5: Number of antibodies in the training/validation/test set of Jin et al. (2021) as well as their subsets of complete complexes.

|  | CDR-H1 | | CDR-H2 | | CDR-H3 | |
| --- | --- | --- | --- | --- | --- | --- |
|  | full | subset | full | subset | full | subset |
| training | 4,050 | 2,131 | 3,876 | 2,035 | 3,896 | 2,051 |
| validation | 359 | 170 | 483 | 241 | 403 | 227 |
| test | 326 | 184 | 376 | 209 | 437 | 207 |
| Total | 4,735 | 2,485 | 4,735 | 2,485 | 4,735 | 3,127 |

## B   ITERATIVE TARGET AUGMENTATION ALGORITHM FOR AFFINITY OPTIMIZATION

Notably, the original algorithm is designed for discrete properties, while in § 4.3 the affinity is continuous. Therefore, we adapt ITA for compatibility with our affinity scorer as follows. The core is that we maintain a list of high-quality candidates for each antibody to be optimized during the ITA process. In each iteration, we produce $C$ candidates for each antibody and sort them together with the candidates in the high-quality list according to the scores. Then we retain the top-$k$ candidates while dropping all others. The above process goes through all the candidates in the current list before entering the next iteration. It is expected that the distribution of the generated antibodies will move towards the higher-affinity landscape. In particular, we run 20 iterations for each pretrained generative model by setting $C = 50$ and $k = 4$.

## C   THE PREDICTOR USED IN AFFINITY OPTIMIZATION

According to Shan et al. (2022), the output of the $\Delta\Delta G$ predictor has been calibrated under the unit of kcal/mol, and the correlation between the predicted $\Delta\Delta G$ and the experimental value is 0.65 on

the test set. It means if the predicted $\Delta\Delta G$ is $x$, then the binding affinity is decreased by $x$ kcal/mol under a correlation of 0.65.

## D EXPERIMENT DETAILS AND HYPERPARAMETERS

For all models incorporating antigen, we select 48 residues closest to the antibody in terms of the alpha carbon distance as the antigen information. We conduct experiments on a machine with 56 CPU cores and 10 GeForce RTX 2080 Ti GPUs. Models using iterative refinement decoding have intensive GPU memory requirements and are therefore trained with the data-parallel framework of Pytorch on 6 GPUs. Other models only need 1 GPU to train. We use Adam optimizer with $lr = 0.001$ and decay the learning rate by 0.95 every epoch. The batch size is set to be 16. All models are trained for 20 epochs and the checkpoint with the lowest loss on the validation set is selected for testing. The training strategy is consistent across different experiments in the paper, and the learning rate of ITA finetuning is also set to 0.001. Furthermore, we provide the hyperparameters for baselines and our MEAN in Table 6. For the RosettaAD (Alford et al., 2017) used in § 4.2, we adopt the denovo design protocol initializing with random CDRs presented in its manual[4].

Table 6: Hyperparameters for each model.

| hyperparameter | value | description |
| --- | --- | --- |
| shared | | |
| vocab size | 25 | There are 21 categories of amino acids in eukaryote, plus the 4 special tokens for heavy chain, light chain, antigen and mask. |
| dropout | 0.1 | Dropout rate. |
| (C-)LSTM | | |
| hidden_size | 256 | Size of the hidden states in LSTM. |
| n_layers | 2 | Number of layers of LSTM. |
| (C-)RefineGNN | | |
| hidden_size | 256 | Size of the hidden states in the message passing network (MPN). |
| k_neighbors | 9 | Number of neighbors to include in k-nearest neighbor (KNN) graph. |
| block_size | 4 | Number of residues in a block. |
| n_layers | 4 | Number of layers in MPN. |
| num_rbf | 16 | Number of RBF kernels to consider. |
| MEAN | | |
| embed_size | 64 | Size of trainable embeddings for each type of amino acids. |
| hidden_size | 128 | Size of hidden states in MEAN. |
| n_layers | 3 | Number of layers in MEAN. |
| alpha | 0.8 | The weight to balance sequence loss and structure loss. |
| n_iter | 3 | Number of iterations for progressive full-shot decoding. |

For training our MEAN on the split in RefineGNN paper setting, we set $alpha = 0.6$, batch size to be 8, and total training epochs to be 30.

## E PROOF OF THEOREM 1

Our MEAN satisfies E(3)-equivariance as demonstrated in Theorem 1:

**Theorem 1.** *We denote the translation process by MEAN as $\{(p_i, \tilde{\boldsymbol{Z}}_i)\}_{i \in \mathcal{V}_C} = f(\{h_i^{(0)}, \boldsymbol{Z}_i^{(0)}\}_{i \in \mathcal{V}})$, then $f$ is E(3)-equivariant. In other words, for each transformation $g \in E(3)$, we have $f(\{h_i^{(0)}, g \cdot \boldsymbol{Z}_i^{(0)}\}_{i \in \mathcal{V}}) = g \cdot f(\{h_i^{(0)}, \boldsymbol{Z}_i^{(0)}\}_{i \in \mathcal{V}})$, where the group action $\cdot$ is instantiated as $g \cdot \boldsymbol{Z} := \boldsymbol{O}\boldsymbol{Z}$ for orthogonal transformation $\boldsymbol{O} \in \mathbb{R}^{3 \times 3}$ and $g \cdot \boldsymbol{Z} := \boldsymbol{Z} + \boldsymbol{t}$ for translation transformation $\boldsymbol{t} \in \mathbb{R}^3$.*

In the following, we denote $f'(\boldsymbol{Z}) = \boldsymbol{Z}^T\boldsymbol{Z}/|\boldsymbol{Z}^T\boldsymbol{Z}|_F$ and the orthogonal group as $O(3) = \{\boldsymbol{O} \in \mathbb{R}^{3 \times 3} | \boldsymbol{O}^T\boldsymbol{O} = \boldsymbol{O}\boldsymbol{O}^T = \boldsymbol{I}_d\}$. Prior to the proof, we first present the necessary lemmas below:

---

[4]https://www.rosettacommons.org/docs/latest/application_documentation/antibody/RosettaAntibodyDesign

**Lemma 1.** *The function $f' : \mathbb{R}^{3 \times m} \to \mathbb{R}^{m \times m}$ is invariant on O(3). Namely, $\forall g \in O(3)$, we have $f'(g \cdot \mathbf{Z}) = f'(\mathbf{Z})$, where $g \cdot \mathbf{Z} := \mathbf{OZ}$.*

*Proof.* $\forall g \in O(3)$, we have $g \cdot \mathbf{Z} := \mathbf{OZ}$. Therefore, we can derive the following equations:

$$f'(g \cdot \mathbf{Z}) = \frac{(\mathbf{OZ})^T (\mathbf{OZ})}{|(\mathbf{OZ})^T (\mathbf{OZ})|_F} = \frac{\mathbf{Z}^T \mathbf{O}^T \mathbf{OZ}}{|\mathbf{Z}^T \mathbf{O}^T \mathbf{OZ}|_F} \tag{9}$$

$$= \frac{\mathbf{Z}^T \mathbf{Z}}{|\mathbf{Z}^T \mathbf{Z}|_F} = f(\mathbf{Z}), \tag{10}$$

which conclude the invariance of $f'$ on $O(3)$. □

**Lemma 2.** *We denote the internal context encoder of layer $l$ as $\{(h_i^{(l+0.5)}, \mathbf{Z}_i^{(l+0.5)})\}_{i \in \mathcal{V}} = \sigma_{in}(\{(h_i^{(l)}, \mathbf{Z}_i^{(l)})\}_{i \in \mathcal{V}})$, then $\sigma_{in}$ is E(3)-equivariant.*

*Proof.* $\forall g \in E(3)$, we have $g \cdot \mathbf{Z}_i^{(l)} := \mathbf{OZ}_i^{(l)} + \mathbf{t}$ where $\mathbf{O} \in O(3)$ and $\mathbf{t} \in \mathbb{R}^3$. Therefore we have the relative coordinates after transformation as $g \cdot \mathbf{Z}_i^{(l)} - g \cdot \mathbf{Z}_j^{(l)} = \mathbf{OZ}_{ij}^{(l)}$. According to Lemma 1 , during the propagations of our internal context encoder, we have $E(3)$-invariant messages:

$$m_{ij} = \phi_m(h_i^{(l)}, h_j^{(l)}, f'(\mathbf{OZ}_{ij}^{(l)}), e_{ij}) = \phi_m(h_i^{(l)}, h_j^{(l)}, f'(\mathbf{Z}_{ij}^{(l)}), e_{ij}), \tag{11}$$

Then it is easy to derive that the hidden states are $E(3)$-invariant and the coordinates are $E(3)$-equivariant as follows:

$$h_i^{(l+0.5)} = \phi_h(h_i^{(l)}, \sum_{j \in \mathcal{N}(i|\mathcal{E}_{in})} m_{ij}), \tag{12}$$

$$\mathbf{OZ}_i^{(l+0.5)} + \mathbf{t} = \mathbf{O}[\mathbf{Z}_i^{(l)} + \frac{1}{|\mathcal{N}(i|\mathcal{E}_{in})|} \sum_{j \in \mathcal{N}(i|\mathcal{E}_{in})} \mathbf{Z}_{ij}^{(l)} \phi_Z(m_{ij})] + \mathbf{t} \tag{13}$$

$$= (\mathbf{OZ}_i^{(l)} + \mathbf{t}) + \frac{1}{|\mathcal{N}(i|\mathcal{E}_{in})|} \sum_{j \in \mathcal{N}(i|\mathcal{E}_{in})} (\mathbf{OZ}_{ij}^{(l)}) \phi_Z(m_{ij}), \tag{14}$$

Therefore we have $g \cdot \sigma_{in}(\{(h_i^{(l)}, \mathbf{Z}_i^{(l)})\}_{i \in \mathcal{V}}) = \sigma_{in}(\{(h_i^{(l)}, g \cdot \mathbf{Z}_i^{(l)})\}_{i \in \mathcal{V}})$. □

**Lemma 3.** *We denote the external attentive encoder as $\{(h_i^{(l+1)}, \mathbf{Z}_i^{(l+1)})\}_{i \in \mathcal{V}} = \sigma_{ex}(\{(h_i^{(l+0.5)}, \mathbf{Z}_i^{(l+0.5)})\}_{i \in \mathcal{V}})$, then $\sigma_{ex}$ is E(3)-equivariant.*

*Proof.* Similar to the proof of Lemma 2, we first derive that the query, key and value vectors are $E(3)$-invariant:

$$q_i = \phi_q(h_i^{(l+0.5)}), \tag{15}$$

$$k_{ij} = \phi_k(f'(\mathbf{OZ}_{ij}^{(l+0.5)}), h_j^{(l+0.5)}) = \phi_k(f'(\mathbf{Z}_{ij}^{(l+0.5)}), h_j^{(l+0.5)}), \tag{16}$$

$$v_{ij} = \phi_v(f'(\mathbf{OZ}_{ij}^{(l+0.5)}), h_j^{(l+0.5)}) = \phi_v(f'(\mathbf{Z}_{ij}^{(l+0.5)}), h_j^{(l+0.5)}), \tag{17}$$

which directly lead to the $E(3)$-invariance of attention weights $\alpha_{ij} = \frac{\exp(q_i^\top k_{ij})}{\sum_{j \in \mathcal{N}(i|\mathcal{E}_{ex})} \exp(q_i^\top k_{ij})}$.

Again it is easy to derive that the hidden states are $E(3)$-invariant and the coordinates are $E(3)$-equivariant as follows:

$$h_i^{(l+1)} = h_i^{(l+0.5)} + \sum_{j \in \mathcal{N}(i|\mathcal{E}_{ex})} \alpha_{ij} v_{ij}, \tag{18}$$

$$\mathbf{OZ}_i^{(l+1)} + \mathbf{t} = \mathbf{O}(\mathbf{Z}_i^{(l+0.5)} + \sum_{j \in \mathcal{N}(i|\mathcal{E}_{ex})} \alpha_{ij} \mathbf{Z}_{ij}^{(l+0.5)} v_{ij}) + \mathbf{t} \tag{19}$$

$$= (\mathbf{OZ}_i^{(l+0.5)} + \mathbf{t}) + \sum_{j \in \mathcal{N}(i|\mathcal{E}_{ex})} \alpha_{ij} \mathbf{OZ}_{ij}^{(l+0.5)} v_{ij}. \tag{20}$$

Therefore we have $g \cdot \sigma_{ex}(\{(h_i^{(l+0.5)}, \mathbf{Z}_i^{(l+0.5)})\}_{i \in \mathcal{V}}) = \sigma_{ex}(\{(h_i^{(l+0.5)}, g \cdot \mathbf{Z}_i^{(l+0.5)})\}_{i \in \mathcal{V}})$. □

With the above lemmas, we are ready to present the full proof of Theorem 1 as follows:

*Proof.* For each layer in MEAN, according to Lemma 2 and 3, we have:

$$\{(h_i^{(l+1)}, g \cdot \boldsymbol{Z}_i^{(l+1))}\}_{i \in \mathcal{V}} = \sigma_{\text{ex}}(\{(h_i^{(l+0.5)}, g \cdot \boldsymbol{Z}_i^{(l+0.5)})\}_{i \in \mathcal{V}}) = \sigma_{\text{ex}}(\sigma_{\text{in}}(\{(h_i^{(l)}, g \cdot \boldsymbol{Z}_i^{(l)})\}_{i \in \mathcal{V}})), \tag{21}$$

Since $p_i$ is obtained by applying softmax on the hidden representations from the output module, which shares the same formula as $\sigma_{\text{in}}$, and $\tilde{\boldsymbol{Z}}_i$ is the same as the coordination from the output module, it is easy to derive:

$$\{(p_i, g \cdot \tilde{\boldsymbol{Z}}_i)\}_{i \in \mathcal{V}} = f(\{h_i^{(0)}, g \cdot \boldsymbol{Z}_i^{(0)}\}_{i \in \mathcal{V}}), \tag{22}$$

$$g \cdot f(\{h_i^{(0)}, \boldsymbol{Z}_i^{(0)}\}_{i \in \mathcal{V}}) = \{(p_i, g \cdot \tilde{\boldsymbol{Z}}_i)\}_{i \in \mathcal{V}}, \tag{23}$$

Therefore we have:

$$f(\{h_i^{(0)}, g \cdot \boldsymbol{Z}_i^{(0)}\}_{i \in \mathcal{V}}) = g \cdot f(\{h_i^{(0)}, \boldsymbol{Z}_i^{(0)}\}_{i \in \mathcal{V}}), \tag{24}$$

which concludes Theorem 1. $\qquad\square$

## F    HUBER LOSS

We use Huber loss (Huber, 1992) for the modeling of coordinations, which is defined as follows:

$$l(x, y) \;=\; \left\{ \begin{array}{l} 0.5\,(x-y)^2,\; if\; |x-y| \;<\; \delta, \\ \delta \;\cdot\; (|x-y| \;-\; 0.5 \;\cdot\; \delta),\; else \end{array} \right. \tag{25}$$

It reads that if the L1 norm of $|x - y|$ is smaller than $\delta$, it is MSE loss, otherwise it is L1 loss. At the beginning of the training, the deviation of the predicted structure and ground truth is large and the L1 term makes the loss less sensitive to outliers than MSE loss. When the training is almost done, the deviation is small and the MSE loss provides smoothness near 0. In practice, we find that directly using MSE loss occasionally causes NaN at the beginning of the training while Huber loss leads to a more stable training procedure. We set $\delta = 1$ in our experiments.

## G    COMPLEXITY ANALYSIS

We provide the complexity analysis here. Suppose the numbers of nodes in the antigen, the light chain, and the heavy chain are $N_A$, $N_L$, and $N_H$, respectively. The message passing in Eq. (1-6) for each node involves $K$ neighbors at maximum, and it is performed over $L$ rounds at each iteration. Then, the complexity of our algorithm over $T$ iterations becomes $O(2LKT(N_A + N_L + N_H))$, where the number 2 refers to the joint computation of the internal and external encoders. Similarly, the complexity of RefineGNN with only heavy chain is $O(LKT'N_H)$, where $T'$ denotes the number of iterations, namely, the length of the CDR region. We would like to specify these two points:

1. Even if our algorithm contains more nodes than RefineGNN ($(N_A + N_L + N_H)$ vs $N_H$), the extra computation overhead does not remarkably count, since the message passing for each node can be parallelly computed in current deep learning platforms (e.g. Pytorch).

2. We apply the full-shot decoding other than the autoregressive mechanism used in RefineGNN, hence $T$ ($T = 3$ in our experiments) is much smaller than $T'$ (usually larger than 10), implying more efficiency of our method.

## H    MODELING WITH RANDOMNESS

The current model is deterministic given the same inputs, but in some scenarios, the diversity of samples is required. To tackle this, we inject randomness into our model by adding standard Gaussian noises to the initialized coordinates. We denote this model as rand-MEAN and evaluate it on the tasks of sequence-structure modeling and antigen-binding CDR-H3 design. The results in Table 7 and Table 8 suggest injecting randomness into MEAN has acceptable impacts on the performance.

Table 7: 10-fold cross validation mean (standard deviation) for 1D sequence and 3D structure modeling on SAbDab (§4.1) of rand-MEAN.

| Model | CDR-H1 | | CDR-H2 | | CDR-H3 | |
|---|---|---|---|---|---|---|
| | AAR | RMSD | AAR | RMSD | AAR | RMSD |
| MEAN | **58.29±7.27%** | **0.98±0.16** | **47.15±3.09%** | **0.95±0.05** | **36.38±3.08%** | **2.21±0.16** |
| rand-MEAN | 56.50±6.44% | **0.98±0.12** | 44.01±2.72% | 1.18±0.20 | 35.68±2.29% | 2.36±0.19 |

Table 8: Amino acid recovery (AAR), TM-score and RMSD for CDR-H3 design on RAbD benchmark of rand-MEAN.

| Model | AAR | TM-score | RMSD |
|---|---|---|---|
| MEAN | 36.77% | **0.9812** | **1.81** |
| rand-MEAN | **37.30%** | 0.9794 | **1.81** |

## I ATTENTION VISUALIZATION

In the external attentive encoder, we apply the attention mechanism to evaluate the weights between residues in different components. It will be interesting to visualize what patterns these attentions will discover. For this purpose, we extract the attention weights between the antibody and the antigen from the last layer, and check if they can reflect the binding energy calculated by Rosetta (Alford et al., 2017). In detail, for each residue in CDR-H3, we first identify the residue in the antigen that contributes the most to its binding energy. Then we calculate the rank of the identified residue according to the attention weights yielded by MEAN. We obtain the relative rank by normalizing it with the total number of antigen residues in the interface. If the attention weights are meaningful, then the resultant rank distribution will be bias towards small numbers; otherwise, they are distributed evenly between 0 and 1. Excitingly, Figure 5 (B) displays that we arrive at the former case, indicating the close correlation between our attention weights and the binding energy calculated by Rosetta. Figure 5 (A) also visualizes an example of attention weights and the corresponding energy map, which shows that their distributions are similar.

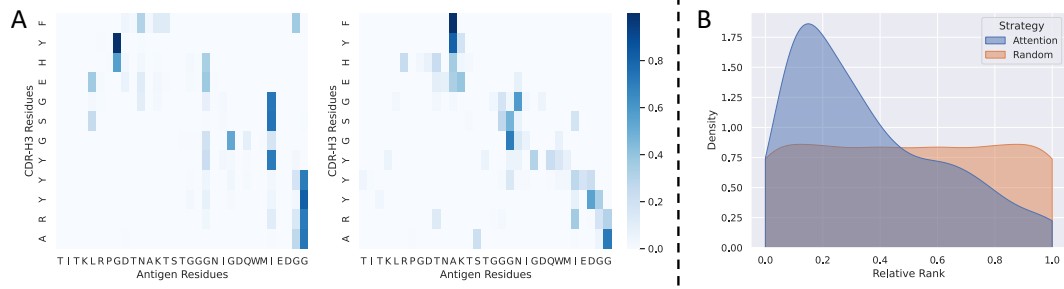

Figure 5: (**A**) The left is the attention weights from the residues in CDR-H3 to those in antigen (PDB: 4ydk). The right is the relative energy contribution of each pair of residues calculated by Rosetta. (**B**) The density maps of the relative ranks for the most contributing residue pairs.

## J HOW GRAPH CHANGES DURING PROGRESSIVE DECODING

We depict the variations of the density distribution of PPL and RMSD across different rounds in the progressive full-shot decoding in Figure 6. Between different rounds, the distribution of PPL remains similar, which is expected since we exert supervision with the ground truth in all the rounds. It is beyond expectation that even if we only supervise the predicted coordination of the last round, the distribution of RMSD shifts rapidly towards the optimal direction.

We additionally calculate the recovery rate of edges in the ground truth graph on the test set, in terms of internal edges within each component and external edges across different components. For the internal edges, the recovery rate is 89% in the beginning and 95% in the end; for the external edges, the recovery rate is 11% in the beginning and 84% in the end. The results above indicate that the linear initialization is able to recover a large part of the internal edges but only a very small

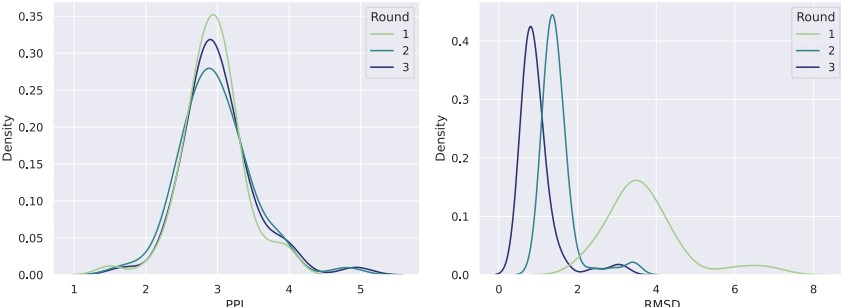

Figure 6: The density of PPL and RMSD of different rounds in progressive full-shot decoding. The area under each curve integrates to 1.

percentage of external edges. By our model, we can predict a major part of both internal and external edges, suggesting the validity of our design.

## K  LOCAL GEOMETRY

Since both RMSD and TM-score reflect the correctness of global geometry, we further provide RMSD of bond lengths and angles to validate the local geometry for Section 4.2. The bond lengths are measured in angstroms and the angles consist of the three conventional dihedral angles of the backbone structure, namely $\phi, \psi, \omega$ (Ramachandran et al., 1963). The RMSD of angles is implemented as the average of RMSD on their cosine values. The results shown in Table 9 indicate that our model still achieves much better performance regarding the local geometry. We find that RefineGNN achieves relatively low performance on modeling local geometry, which might be because that its indirect loss on various invariant features cannot ensure atoms in the backbone are equally supervised. For example, the precision of the coordinates of the carboxy carbon is relatively low with a high RMSD of 10.8, which results in the high error in bond lengths of local geometry.

Table 9: RMSD of bond lengths and dihedral angles of the backbone structure.

| Model | Bond lengths | Angles($\phi, \psi, \omega$) |
|---|---|---|
| RefineGNN | 10.56 | 0.456 |
| C-RefineGNN | 9.49 | 0.423 |
| MEAN | **0.37** | **0.302** |

## L  FULL ANTIGEN OR ONLY EPITOPE

In this paper, we use the 48 residues closest to the antibody to represent the antigen information. From the perspective of biology, these residues compose the epitope (i.e. the binding position of antibodies) which is usually located by a biological expert or detected via certain computational methods (Haste Andersen et al., 2006) beforehand, and they usually provide enough information for designing the antigen-binding antibodies. Theoretically, only the residues close to the antibody will affect the message passing between the antigen and the antibody because we construct edges based on a cutoff of $C_\alpha$ distance, therefore the performance should be similar regardless of using the full antigen or only the epitope. Practically, we also explore the influence of exclusion of other residues in the antigen on our model. Specifically, we incorporate full antigen in the experiment of § 4.2 and present the results in Table 10. As expected, incorporating full antigen results in similar performance to the epitope-only strategy with slight change in AAR and structure modeling.

Table 10: Results of antigen-binding CDR-H3 design given different forms of antigen information.

| Antigen | AAR | TM-score | RMSD |
|---|---|---|---|
| epitope-only | 36.77% | **0.9812** | **1.81** |
| full | **37.92%** | 0.9798 | 2.00 |

## M    LIMITATIONS AND FUTURE WORK

**Sidechain generation**    In this paper, we follow the previous settings in Jin et al. (2021) and only model the backbone geometry for fair comparisons. However, sidechains also play an essential role in the interactions between antigens and antibodies. A potential method is to extend our method to incorporate sidechains by replacing the current node feature with the local full-atom graph of each residue. We leave this for future work.

**Data augmentation**    The size of existing data of antibodies is still small since it is hard to obtain the 3D structures of antibodies in practice. There are several possible ways to do data augmentation for further improvement. The first is to leverage pretrained models on sequences. Since the 1D sequence data of antibodies are much more abundant than 3D data, it is possible to pretrain an embedding model based on 1D sequences and inject the pretrained model into our framework. It is also promising to conduct the pretraining on general protein data from PDB and then carry out finetuning on the antibody dataset. Another potential way is to use Alphafold to produce pseudo 3D structures from 1D sequences for pretraining. We leave the above considerations for future work.

## N    EXAMPLES

We display more examples of antigen-binding CDR-H3 designed by our MEAN in Figure 7.

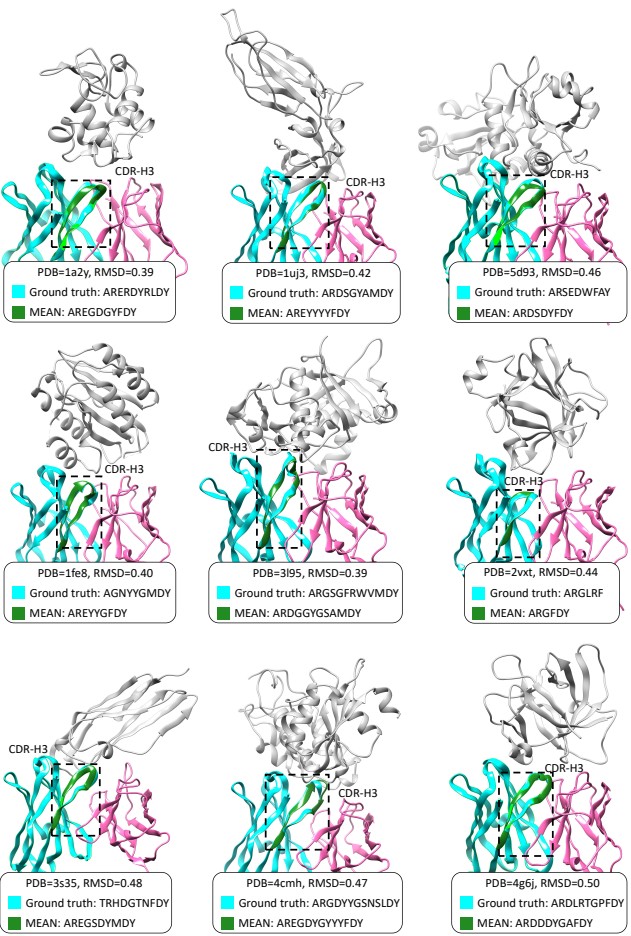

Figure 7: More examples from antigen-binding CDR-H3 design by our MEAN.

## O    TOWARDS THE REAL-WORLD QUESTION

Note that the efficacy of our pipeline for affinity optimization is influenced by the generalizability of the predictor, hence how to choose a desirable predictor is vital. As proof of concept, we currently apply the predictor in Shan et al. (2022) for its easy implementation and fast computation. Since our pipeline is general, it is possible to replace the predictor with other variants, such as wetlab validation, which can return real affinity but is time-consuming. One can also combine the advantages of the learning-based predictor and wetlab validation to improve the generalizability while keeping efficiency, by, for example, choosing only top-k samples and using the wetlab feedback to rectify the predictor, creating a so-called closed loop between "dry computation" and "wet experiment" akin to the pipeline used in Shan et al. (2022). As increasing attention has been paid to this domain, we believe more and more robust and efficient predictors will emerge in the future.

While the pipeline for affinity optimization only addresses a narrow need in the field of antibody discovery, we also discuss the potential pipeline for the 'real-world' question here (i.e. generate a binidng antibody given an arbitrary antigen). The 'real-world' question might be decomposed into several components: epitope identification, antibody structure prediction, docking, CDR design, and affinity prediction. Each component itself is challenging and currently a promising topic in the community. In § 4.4, we actually combined the last three components, where we used HDock for global docking to form the initial complex, our MEAN for CDR design, and Rosetta for affinity computation. If we further add the components for epitope identification and antibody structure prediction (such as Igfold (Ruffolo et al., 2021; 2022)), we are able to set up an end2end pipeline for antibody discovery: it can output a desirable antibody (1D sequence and 3D structure) for any given antigen target. However, setting up such an end2end pipeline is challenging, as the accumulated errors from the former components will easily make the latter fail. A potential solution is making all components learnable, and tuning them as a whole.

Lastly, our proposed model and the proof-of-concept experiments we implemented will provide valuable clues for future exploration to derive enhanced techniques. With the efforts of all researchers in the field, we have reason to believe that this ultimate problem will be solved, perhaps step by step.

