# OpenReview forum: "Conditional Antibody Design as 3D Equivariant Graph Translation"
_ICLR.cc/2023/Conference — ICLR 2023 notable top 5%_

### Official Review · Reviewer_tzFw · 2022-10-22

**Confidence:** 4
**Correctness:** 4
**Technical Novelty And Significance:** 3
**Empirical Novelty And Significance:** 4
**Recommendation:** 8

**Clarity, Quality, Novelty And Reproducibility:**

Clarity: The paper is mostly well written. The experiment section is perhaps a little dense and a few more signposts could be provided for easier reading, and easier understanding what each experiment is supposed to show.

Quality: The paper is technically sound, with thorough benchmarks and baselines. Methods are well explained. Results are quite good.

Novelty: While some components have been introduced in previous papers, the authors combine them in a novel way, introduce some new modeling ideas, and achieve excellent results.

Reproducibility: Authors provided anonymized source code and thoroughly documented all dataset generation methods.

**Strength And Weaknesses:**

Strengths:
* Multiple novel contributions.
    - Modeling all components of the antibody/antigen complex
    - joint sequence/structure design
    - iterative prediction of model outputs.
* Convincing and strong results on multiple downstream benchmarks. Dataset splits are constructed well and baselines (both neural and non-neural) are compared to appropriately.
* Ablations of model are included to verify claims as to the utility of the different additions.
* Well written paper. Model and experimental results are well explained and an appropriate level of detail is provided to understand the data split strategy and baselines.

Weaknesses:
To be honest, I see no major weaknesses in the paper that must be addressed before publication. Below, I've listed some questions for future work and other minor comments.

Questions / Future Work:
- It is interesting that the graph is constructed without full knowledge of the location of all residues - a linear imputation is used for missing residues. It would be nice to analyze this a bit further - how much do residues move from this initialization? How "accurate" is the initial graph structure - are there edges missing/present in the initial graph vs. in the "true" graph and how much does the graph change over iterations of the iterative refinement?
- Related to the point above, it seems this constructions works in part because much of the antibody structure is already known. Would it be possible to apply the method to general sequence design, as opposed to loop re-design (e.g. Dauparas et al. 2022, science.org/doi/10.1126/science.add2187)? This would be a breakthrough result if so.
- Ultimately, in-silico evaluations for protein design only go so far. It would be great to see the designs verified experimentally.
- The authors note the current model cannot model sidechains. I do think this is a limitation that future work should explore, as it seems this would be important for the design of binding.

Minor Comments:
- It was slightly hard for me to associate different tables with experiments / datasets at a glance (e.g. it took me a little bit to figure out what exactly is the difference between Table 1 and Table 2 (left)). Perhaps the experiments section could include a bulleted list of datasets used, or this could be more clearly highlighted in the table captions.

**Summary Of The Paper:**

The paper proposes a novel method for joint antibody sequence/structure design. The paper introduces / combines multiple innovations:

* Modeling antibody heavy + light chain, along with antigen
* Joint sequence / structure design
* Iterative prediction of outputs

While some of these have been introduced before, the combination is very interesting and results in compelling results on the downstream task of antibody design.

**Summary Of The Review:**

This is a good paper that is a clear candidate for acceptance at ICLR. Methodology is novel, benchmarks are thorough, and results are good. The chosen problems are highly important and have potential for major impact.

---

> ### Author Response · Authors · 2022-11-12
> **Thank you for your review**
>
> Thank you for your constructive comments!
>
> > Q1: It is interesting that the graph is constructed without full knowledge of the location of all residues - a linear imputation is used for missing residues. It would be nice to analyze this a bit further - how much do residues move from this initialization? How "accurate" is the initial graph structure - are there edges missing/present in the initial graph vs. in the "true" graph and how much does the graph change over iterations of the iterative refinement?
>
> Nice suggestions! About "how much do residues move from this initialization", we actually provided the change of RMSD across different rounds in Figure 6 of Appendix I, which somehow explains the position movement of residues from the linear initialization. It shows remarkable shifts occurring from the first to the second round, and fine-grained adjustment from the second to the third round. We have added the reference to Figure 6 in the revision.
> To answer "how does the graph chage", we  additionally calculate the recovery rate of edges in the ground truth graph on the test set, in terms of internal edges within each component and external edges accross different components. For the internal edges, the recovery rate is 89% in the beginning and 95% in the end;  for the external edges, the recovery rate is 11% in the beginning and 84% in the end. The results above indicate that the linear initialization is able to recover a large part of the internal edges but only a very small percentage of external edges. By our model, we can predict a major part of both internal and external edges, suggesting the validity of our design. We have added the above discussions in Appendix I.
>
> > Q2: Related to the point above, it seems this constructions works in part because much of the antibody structure is already known. Would it be possible to apply the method to general sequence design, as opposed to loop re-design (e.g. Dauparas et al. 2022, science.org/doi/10.1126/science.add2187)? This would be a breakthrough result if so.
>
> Thank you for the great suggestion! Here, we formulate an inverse folding task on antibodies, by inputting the 3D structures and outputting the 1D sequences, similar to the design in Dauparas et al. 2022. Our model MEAN is still applicable in this case, by simply changing the input and output vectors. Interestingly, we find that the average amino acid recovery rate is  73.5% on the test set. This validates that our model somehow has the potential to do inverse folding of native proteins (including multimers). While inverse folding in general domain requires extra datasets and  other careful considerations, we are eager to try it out in future work!
>
> > Q3: Ultimately, in-silico evaluations for protein design only go so far. It would be great to see the designs verified experimentally.
>
> Thank you for the comment. We are actually working on the wet-experimental validation. We have optimized an existing antibody binding to a Histone-modifying peptide, and the wetlab validation of the binding affinity is in progress. We are willing to include these experimental results in future work.
>
> > Q4: The authors note the current model cannot model sidechains. I do think this is a limitation that future work should explore, as it seems this would be important for the design of binding.
>
> Great suggestion! We also believe this is a promising direction. Actually, our multi-channel message passing can be directly extended to the modeling of sidechains by simply adding more channels. However, the number of atoms in each residue varies according to its amino acid type, which needs to be carefully considered. We had better leave it for future exploration.
>
> > Q5: It was slightly hard for me to associate different tables with experiments / datasets at a glance (e.g. it took me a little bit to figure out what exactly is the difference between Table 1 and Table 2 (left)). Perhaps the experiments section could include a bulleted list of datasets used, or this could be more clearly highlighted in the table captions.
>
> Sorry for the confusion. Per your suggestion, we have added the association with the experiments (e.g. section number and benchmark) in each table caption in the revision.

---

### Official Review · Reviewer_n2oD · 2022-10-23

**Confidence:** 4
**Correctness:** 3
**Technical Novelty And Significance:** 4
**Empirical Novelty And Significance:** 3
**Recommendation:** 8

**Clarity, Quality, Novelty And Reproducibility:**

The paper is written quite clearly. The ideas are laid out well with clear demarcation of where the authors are expanding on the limits of the prior papers that they cite. The method is written out in detail displaying the most important aspects of the architecture. The proof of equivariance is easy to follow. The most unclear aspect of the method was the use of  si  in the progressive full short decoding. I understand these to be the learned embeddings, but they appear to be different from the hi outputted by the last layer of the model. Furthermore, as mentioned above, the authors could have discussed more their reasoning behind different message passing and/or attention paradigms beyond the equivariant construction.

In terms of novelty, while the paper is addressing a question that has been attempted by previous papers, it offers noteworthy contributions in the modelling approach. The authors propose an equivariant model over the invariance offered by RefineGNN (the primary benchmark). There have been many methods that have offered equivariant message passing schemes (eg. Jing 2020, Fuchs 2022). However, the specific construction of the update layers is novel as far as this paper is concerned, and the utilization of equivariance in the antibody sequence-structure co-design problem is novel.

They also try to extend their method to conditions in which a binding complex is not available by leveraging docking. The soundness of this approach is conditional on how believable the docking results are. However, not having a whole binding complex structure is a more real-world situation, so these developments are a step in the right direction.

The paper contributes all their code for their model along with reproduced benchmarks models, so the results should be highly reproducible.


**Strength And Weaknesses:**

The model architecture the authors develop is very thoughtfully constructed and the gains are demonstrated in the state-of-the-art performance. Their ablation study is an important addition, as it shows which aspect of the method contribute most to its performance gains. The use of Progressive Full-shot decoding was the biggest contributor to performance while also having major run-time gains. This suggests that it could have promise for utilization in other methods as well.

The authors did not give much explanation as to why a message propagation paradigm was used with the internal encoder and a graph attention scheme used in the external encoder. The ablation study also demonstrated that the attention scheme did not contribute hugely to the performance. Some more explication of the modelling decisions would be beneficial.

The “Affinity Optimization” experiment is a great section of the paper, and moves towards real-world usage of these methods. The authors show their method is able to generate a CDR with the biggest decrease in delta-delta-G (a proxy for binding affinity) compared to benchmarks. To make the result more biologically interpretable it would have been useful to have an idea of how much a delta-delta-G decrease contributes to a decrease in binding affinity (as measured by Kd for example). This could have been done using the Shan et al model and comparing to test performance on the SKEMPI set. While this correlation would be different for different protein interactions, it could have been a useful discussion point. Using a general delta-delta-G predictor over an oracle model trained on experimental data is a clever extension ofhe Iterative Target Augmentation algorithm for this use-case. However, while the authors do mention that they only benchmark on antibody complexes within the training set of the Shan et al model to ensure generalizability, this points to how the efficacy of this pipeline for any candidate antibody is limited by the generalizability of the predictor. While this is not a fault of their proposed model, if this pipeline is a major point of utility of this paper, it is a point worth discussing.

While the authors do cite a major paper in the field for sequence-structure co-design (Jin et al 2021), they do not mention the new paper from that group (Jin et al 2022) which makes similar developments on its prior model as this paper performs (eg. modelling the antigen structure in the CDR generation). Furthermore, in terms of benchmarks, while the authors’ model did show state-of-the-art performance when compared to any cited paper, there are some discrepancies in RefineGNN’s RAbD benchmark performance between the original Jin 2021 paper and this paper’s reproduction of it. Lastly, it seems that the authors criticize RefineGNN for not being invariant in its loss calculation, however this is not the case.

A high level thought : An overall weakness of this work is common to much of the published work in this field; i.e.  the work does not  really address the "real world" problem of antibody design; - the 'real-world' question should be given an antigen target, design an antibody that will bind with high affinity and specificity. However good a method is at predicting antibody binding to a specific epitope on a target antigen, and however well the model can generate sequences and structures that bind that epitope - the 'real-world' question should be able to find new epitopes, given an antigen target, design an antibody that will bind with high affinity and specificity.  Nevertheless , these authors do not claim to do this so thats fine - but the ML community should be aware that these evaluations and target problems only address a very narrow need- e.g. I have one epitope  on an antigen of know. structure - can I make a 'better' antibody to that specific target surface?




**Summary Of The Paper:**

Designing Antibody CDR sequences is of great interest for studying antibody biology and for therapeutics research. This task has been addressed computationally using search based frameworks (eg. using rosetta energies) and more recently, machine learning based generative models. This paper introduces a novel equivariant GNN framework for co-generating CDR sequences and structures. They evaluate the generative capacity of this model using standard benchmarks on sequence and structure generation (i.e Amino Acid Recovery/AAR and Structure RMSD). They demonstrate novel utility of this model in the task of optimizing the binding affinity of a candidate antibody. Lastly, they suggest a pipeline for generating a CDR sequence when the binding complex is unknown using a docking search approach.

A major contribution this paper has made to the field is the capacity to conditionally model the antigen structure in the CDR generation. They implement this via an “external context encoder” attention mechanism which passes information from antigen residues to CDR residues in the hidden layer updates. The architecture for the message passing and attention paradigms are also novel and constructed to be equivariant, which is a required feature for a model operating on coordinate data. As the primary function of antibody CDR regions are to bind to target antigens, it follows that modelling the target structure in the generative model would be important. Thus this is an important contribution.

This paper compares its method to a previously published generative sequence-structure co-design model (RefineGNN from Jin et al 2021), an unconditional sequence model (from Saka et al 2021) and an Rosetta energy based search method (RAbD, Adolf-Bryfogle 2018). They show state-of-the-art performance on AAR and RMSD metrics, with quite significant improvements.

The paper also presents a novel sequence generation scheme, Progressive Full-Shot Decoding, which is faster than prior decoding techniques. They show in their ablation study that the decoding scheme hugely contributes to their performance gains.

While a prior generative model paper (Jin et al 2021)  developed an algorithm for optimizing the binding affinity of antibodies, that method required an antigen specific oracle trained on experimental data. This paper takes the approach a step further by leveraging a delta-delta-G predictor from Shan et al 2022, which allows the algorithm to be used on any antibody-antigen pair for which a structure is available. The authors show their model outperforms other benchmarks using this pipeline in terms of generated antibody with lowest energy.


**Summary Of The Review:**

This paper attempts to address the antibody heavy chain CDR sequence-structure co-design problem, though. They extend prior models on this task by incorporating conditioning on the target antigen and the light chain. The importance of incorporating this information is demonstrated by the major performance gains on established benchmarks. They also propose extended utility of this model in the goal of optimizing an antibody’s affinity and also generating a heavy chain CDR when a full binding complex is not present using docking.

There are some limitations in this paper such as a lack of description of certain modelling decisions and asI wrote above, it does not address the CDR sequence-structure problem in general,  rather only for a specific binding surface which by definition has already had a structure of Ab:Antigen solved.  However, the improvements on benchmarks and new ideas of in silico optimization make this paper a good step in the direction of solving the in silico antibody design problem

---

> ### Author Response · Authors · 2022-11-12
> **Thank you for your review (part 3)**
>
> > Q5: The ML community should be aware that these evaluations and target problems only address a very narrow need. The 'real-world' question should be able to find new epitopes, given an antigen target, design an antibody that will bind with high affinity and specificity.
>
> We strongly agree with you that many related ML papers (incuding ours) only address a small part of the real problems of antibody discovery. Even so, we are willing to share our thoughts with you below on how to put our model into a potentially-complete pipeline to fulfil the real purpose. The 'real-world' question might be decomposed into several components: epitope identification, antibody structure prediction, docking, CDR design, and affinity prediction. Each component itself is challenging and currently a promising topic in the community. In Section 4.4, we actually combined the last three components, where we used HDock for global docking to form the initial complex, our MEAN for CDR design, and Rosetta for affinity computation.  If we further add the components for epitope identification and antibody structure prediction (such as Igfold), we are able to set up an end2end pipeline for antibody discovery: it can output a desirable antibody (1D sequence and 3D structure) for any given antigen target.  However, setting up such an end2end pipeline is challenging, as the accumulated errors from the former components will easily make the latter fail.  A potential solution is making all components learnable, and tuning them as a whole.
> Lastly, our proposed model and the proof-of-concept experiments we implemented will provide valuable clues for future exploration to derive enhanced techniques. With the efforts of all researchers in the field, we have reason to believe that this ultimate problem will be solved, perhaps step by step. We have also added the discussion in Appendix N in the revision to address the reviewer's concerns.
>
> > Q6: The most unclear aspect of the method was the use of si in the progressive full short decoding. I understand these to be the learned embeddings, but they appear to be different from the hi outputted by the last layer of the model.
>
> Sorry for the insufficient clarity. We first use $h_i$outputted by the last layer to obtain a predicted probability of amino acid types of each residue, denoted as $p_i$. Then, for each residue, we update its node feature with a weighted sum of the initial embedding $s_i$, where the weights are given by $p_i$. This weighted sum strategy provides smoother inputs and reduces information loss between different rounds. We have further clarified this point in Section 3.3 in the revision.

---

> ### Author Response · Authors · 2022-11-12
> **Thank you for your review (part 2)**
>
> > Q3: However, while the authors do mention that they only benchmark on antibody complexes within the training set of the Shan et al model to ensure generalizability, this points to how the efficacy of this pipeline for any candidate antibody is limited by the generalizability of the predictor. While this is not a fault of their proposed model, if this pipeline is a major point of utility of this paper, it is a point worth discussing.
>
> Yes, the efficacy of our pipeline is influenced by the generalizability of the predictor, hence how to choose a desirable predictor is vital. As proof of concept, we currently apply the Shan et al predictor for its easy implementation and fast computation. Since our pipeline is general, it is possible to replace the predictor with other variants, such as wetlab validation, which can return real affinity but is time-consuming. One can also combine the advantages of  the learning-based predictor and wetlab validation to improve the generalizability while keeping efficiency, by, for example, choosing only top-k samples and using the wetlab feedback to finetune the predictor, creating a so-called closed loop between "dry computation" and "wet experiment" akin to the pipeline used in Shan et al. As increasing attention has been paid to this domain, we believe more and more robust and efficient predictors will emerge in the future. We have added the above discussion in Appendix N in the revision to address the reviewer's concern.
>
> > Q4: (baselines) While the authors do cite a major paper in the field for sequence-structure co-design (Jin et al 2021), they do not mention the new paper from that group (Jin et al 2022) which makes similar developments on its prior model as this paper performs (eg. modelling the antigen structure in the CDR generation). Furthermore, in terms of benchmarks, while the authors’ model did show state-of-the-art performance when compared to any cited paper, there are some discrepancies in RefineGNN’s RAbD benchmark performance between the original Jin 2021 paper and this paper’s reproduction of it. Lastly, it seems that the authors criticize RefineGNN for not being invariant in its loss calculation, however this is not the case.
>
> Thank you for raising the reference to the new paper HERN (Jin et al 2022). While HERN also involves the antigen structure (i.e. epitope), it focuses only on peptide (i.e. CDR-H3) generation, without considering the framework regions of the antibody. Such incomplete modeling will possibly make the generated peptide unable to conform to the geometric constraints of the framework. We have mentioned the comparison with HERN in Related Work in the revision.
>
> As for RefineGNN, we used its officially-released codes in our experiments. The discrepancies in performance are mainly due to different dataset and different train-test split we used. We selected data samples with complete structure (Antigen & Antibody) from SAbDab, leading to 52% of the dataset applied in the original paper of RefineGNN. For fair and reliable comparisons, we trained and evaluated all models over 10-fold cross validation of the same extracted dataset. The results are shown in the top subtable of Table 1. Moreover, in the bottom subtable of Table 1, we employed the same train-test split as RefineGNN paper but only train our MEAN on the subset with complete structures, and directly borrowed RefineGNN from the officially released checkpoint which was trained on the full set; it shows our model, even with less training samples, can still surpass RefineGNN in terms of AAR and achieve comparable performance in structure modeling, on the extracted test set.
>
> On the loss calculation of RefineGNN, there could be some misunderstanding here. At the end of Section 3.3，we were NOT intended to criticize RefineGNN for not being invariant in its loss calculation. Instead, we accept that the loss of RefineGNN is E(3)-invariance, but it achieves this  through the computation of pair-wise distances and angles, which we think is less efficient than our loss calculation by direct supervision of coordinates.

---

> ### Author Response · Authors · 2022-11-12
> **Thank you for your review (part 1)**
>
> Thank you for your detailed and insightful comments! We provide the following responses to your concerns, and would like to share our thoughts with you on some open and long-term questions.
>
> > Q1: The authors did not give much explanation as to why a message propagation paradigm was used with the internal encoder and a graph attention scheme used in the external encoder. The ablation study also demonstrated that the attention scheme did not contribute hugely to the performance. Some more explication of the modelling decisions would be beneficial.
>
> Sorry for current insufficient explanation. The alternating message passing paradigm is motivated by the biological insight that the external interactions between antibodies and antigens are different from those internal interactions within each heavy/light chain. It is thus natural to model them with different modules. The reason why we choose the attention scheme for external interaction modeling is to dynamically learn the  attention weight for each cross-chain residue pair. Although including the attention scheme might not bring in remarkable performance improvement in some cases, it does provide better interpretability. For example, in Figure 5 of Appendix H, we visualized the learned attention weights, which are closely correlated with the relative energy contribution calculated by Rosetta, helping us locate which external interactions are important. We have contained more explanations in the analysis section in the revision.
>
> > Q2: To make the result more biologically interpretable it would have been useful to have an idea of how much a delta-delta-G decrease contributes to a decrease in binding affinity (as measured by Kd for example). This could have been done using the Shan et al model and comparing to test performance on the SKEMPI set.
>
> We agree with the reviewer on this point. Indeed, this has been done in the reference paper (Shan et al, 2022). According to Shan et al, 2022, the output of the delta-delta-G predictor has been calibrated under the unit of kcal/mol, and the correlation between the predicted delta-delta-G and the experimental value is 0.65 on the test set. It means if the delta-delta-G is x, then the binding affinity is decreased by x kcal/mol under a correlation of 0.65. We have added this clarification to Appendix C.

---

### Official Review · Reviewer_JzYy · 2022-10-23

**Confidence:** 3
**Correctness:** 4
**Technical Novelty And Significance:** 4
**Empirical Novelty And Significance:** 4
**Recommendation:** 8

**Clarity, Quality, Novelty And Reproducibility:**

This paper makes solid and high-quality contributions to the antibody design problem, with significant novelty contributions and excellent empirical studies. The details of the method are well clarified. Also, detailed experimental settings have been provided in the appendix for reproducibility.

**Strength And Weaknesses:**

Strengths:
(+) This paper gives a novel formulation of the antibody design problem by including the target antigen and the light chain of the antibody as conditions and designs a novel network architecture for this problem. The novelty contribution is significant and enlightening.
(+) Comprehensive empirical studies and ablation studies are conducted to demonstrate the advantages of the proposed method over prior methods. The experimental design and results are solid.
(+) The writing of the paper is very clear and well-organized.

Weaknesses:
(-) (minor) In the section 3.3, it is better to provide more explanations or details about Huber loss and its advantages.
(-) (minor) In the end of section 3.1, the antibody design is formulated as a prediction problem, i.e., the output and input are one-to-one. In practice, for a specific antigen, is there only one antibody with a unique stable structure that can bind to it? I will appreciate if the authors can explain why the task is considered a prediction problem rather than a generation problem.

**Summary Of The Paper:**

This work presents a novel method for the co-design of 1D sequence and 3D structures of antibody proteins that can bind to the target antigen. The method proposes to include the target antigen and the light chain of the antibody as additional conditions, and designs a novel E(3)-equivariant network to predict the amino acid types and 3D conformations. Experiments show that the proposed method achieves excellent performance.

**Summary Of The Review:**

Overall, this paper proposes a novel method for the antibody design problem and demonstrates the advantages of it through solid experiments. Considering the high-quality contribution of this work, I vote for acceptance.

---

> ### Author Response · Authors · 2022-11-12
> **Thank you for your review**
>
> Thank you for your valuable review! We provide more explanations to address your concerns as follows:
>
> > Q1: In section 3.3, it is better to provide more explanations or details about Huber loss and its advantages
>
> Thanks! Huber loss is defined by:
>
> $$
> l(x, y) = \\left\\{ \\begin{array}{lr}
> 0.5 (x-y)^2, if |x-y| < \\delta, \\\\
> \\delta * (|x-y| - 0.5 * \\delta), else
> \\end{array}\\right.
> $$
>
> which indeed is MSE loss if the L1 norm of (x-y) is smaller than $\delta$,  otherwise L1 loss. This two-phase design is beneficial. At the beginning of the training process, the error is large and the L1 loss makes the loss less sensitive to outliers, compared to MSE loss (as the gradients of MSE loss includes the (x-y) term). When the training is almost done, the error is small and MSE loss provides smoothness near 0. In practice, we find that directly using MSE loss occasionally leads to NaN at the beginning of the training while Huber loss leads to more stable training. Further information about Huber loss can be found on the Wiki page. We have added this explanation in the revision.
>
> > Q2: In the end of section 3.1, the antibody design is formulated as a prediction problem, i.e., the output and input are one-to-one. In practice, for a specific antigen, is there only one antibody with a unique stable structure that can bind to it? I would appreciate it if the authors can explain why the task is considered a prediction problem rather than a generation problem.
>
> Sorry for the ambiguity caused by the misuse of 'predict' which might be better replaced by 'generate' in the end of section 3.1. We actually consider the task a generation problem where many possible antibodies are allowed to bind to one given antigen.  Different solutions can be sampled from the predicted distribution of our model. The initial attempt of using ‘predict’ refers to that the generated sequence and structure should rely on the given context. We have corrected the presentation in the revision to avoid ambiguity.

---

### Official Review · Reviewer_HYV9 · 2022-10-25

**Confidence:** 3
**Correctness:** 4
**Technical Novelty And Significance:** 3
**Empirical Novelty And Significance:** 3
**Recommendation:** 8

**Clarity, Quality, Novelty And Reproducibility:**

Clarity:

The paper is well written and easy to read with only minor typos.

… while keeps refining … --> while keeping refining would be more appropriate

the symmetry of our 3D word --> world

and a set of coordinates Zi ∈ R 3×m consisting of m backbone atoms.

Looks like Zi is a 3 by m matrix because the coordinates of each node is defined with respect to their backbone atoms. If this is the case please make this clear in Section 3.1.

It was not immediately clear what the l+0.5 superscript implies in equations 2 and 3.

Quality and Novelty:

Thanks to the way node and edge features are obtained the proposed algorithm achieves E(3) equivariance which is proved by a theorem. Although existing E(n) graph neural network is used as a building block the design is improved with an attention mechanism. The model generates fullshot sequences in multiple iterations where the number of iterations used in experiments is only 3, which achieves significantly better efficiency for long CDR sequences compared to autoregressive models. Ablation study clearly demonstrates the incremental improvement due to the attention mechanism, introducing global node, and integrating light chain and antigen to offer context.

Reproducibility:

Code is provided. The results should be reproducible with some effort but no attempt to reproduce them has been made.



**Strength And Weaknesses:**

Strengths
+ Progressive fullshot decoding is more scalable and less prone to accumulated errors compared to autoregressive models that generate amino acids one at a time.
+ Light chain and antigen information can provide additional context and can generate CDRs with high binding affinity.
+ A novel equivariant attention mechanism was developed to better capture interactions




**Summary Of The Paper:**

This paper formulates antibody design as E(3)-equivariant graph translation. Input to the graph contains information about the antigen along with the CDR information in the heavy and light chains.  An E(3) equivariant attention network is developed. This network can output 1D sequence and 3D structure of CDRs.  This network operated in 3D coordinate space while maintaining E(3) equivariance and the full geometry of residues. The learning alternates between an internal context encoder and an external attentive encoder and aims to capture long-range and spatially-complicated interactions. CDRs are generated in multiple rounds but each round updates sequences and structures as a whole. The model is tested on three tasks and compared against four baseline models with promising results.

**Summary Of The Review:**

Although the proposed algorithm uses an existing graph architecture as a backbone, new modifications  seems to  make significant improvements over this existing model. The paper is well-written and theoretically sound with strong experimental evidence.

---

> ### Author Response · Authors · 2022-11-12
> **Thank you for your review**
>
> Thank you for your recognition of our contributions! We have improved the clarity of our paper according to your kind suggestions:
>
> > Q1: Typos.
>
> Thank you for your careful reading. We have corrected the mentioned typos and proofread our paper thoroughly.
>
> > Q2: Looks like  Zi is a 3 by m matrix because the coordinates of each node are defined with respect to their backbone atoms. If this is the case, please make this clear in Section 3.1.
>
> Yes, it is the case, and we have further clarified this point in Section 3.1 in the revision.
>
> > Q3: It was not immediately clear what the l+0.5 superscript implies in equations 2 and 3.
>
> Sorry for the confusion. In our model, each l-th layer consists of  a context encoder and an attentive encoder. To distinguish between these two encoders, we apply l+0.5 to denote the output from the context encoder, which will be further updated by the attentive encoder. We have added the explanation to the revision.

---

### Comment · Program_Chairs · 2023-03-21
**Congratulations on winning the ICLR 2023 Outstanding Paper Honorable Mention!!**

Hello authors,

Big congratulations! For more details about the award process this year, please see https://blog.iclr.cc/2023/03/21/announcing-the-iclr-2023-outstanding-paper-award-recipients/

Best,

Mengdi, on behalf of ICLR 2023 PCs

---

### Decision · Program_Chairs · 2023-01-20

**Decision:**

Accept: notable-top-5%

**Justification For Why Not Higher Score:**

N/A

**Justification For Why Not Lower Score:**

The proposed method was unanimously consider novel and to have great potential.

**Metareview: Summary, Strengths And Weaknesses:**

The paper proposes a technique to co-design the 1D sequence and 3D structure of antibodies by phrasing it as a conditional graph translation problem. The proposed Multi-channel Equivariant Attention Network (MEAN) is novel, and demonstrated to lead to significant performance improvements over current SOTA. The reviewers unanimously and enthusiastically recommend acceptance.

**Note From Pc:**

if the above contains the word "oral" or "spotlight" please see: "oral" presentation means -> notable-top-5% and "spotlight" means -> notable-top-25%. As stated in our emails, we are disassociating presentation type from AC recommendations